# Comparing Predictive Utility of Head Computed Tomography Scan-Based Scoring Systems for Traumatic Brain Injury: A Retrospective Study

**DOI:** 10.3390/brainsci13081145

**Published:** 2023-07-29

**Authors:** Moein Khormali, Saeed Soleimanipour, Vali Baigi, Hassan Ehteram, Hamidreza Talari, Khatereh Naghdi, Omid Ghaemi, Mahdi Sharif-Alhoseini

**Affiliations:** 1Sina Trauma and Surgery Research Center, Tehran University of Medical Sciences, Tehran 14166-34793, Iran; moeinkhormali@gmail.com (M.K.); v-baigi@sina.tums.ac.ir (V.B.); khaterehnaqdi@gmail.com (K.N.); 2Department of Radiology, Sina Hospital, Tehran University of Medical Sciences, Tehran 14166-34793, Iran; saeedmd2014@gmail.com; 3Department of Epidemiology and Biostatistics, School of Public Health, Tehran University of Medical Sciences, Tehran 14166-34793, Iran; 4Department of Pathology, School of Medicine, Kashan University of Medical Sciences, Kashan 87159-88141, Iran; ehteram.kaums@gmail.com; 5Trauma Research Center, Kashan University of Medical Sciences, Kashan 87159-88141, Iran; talari2008hr@yahoo.com; 6Department of Radiology, Kashan University of Medical Sciences, Kashan 87159-88141, Iran; 7Department of Radiology, Imam Khomeini Hospital, Tehran University of Medical Science, Tehran 14166-34793, Iran; omid_gh62@yahoo.com; 8Department of Radiology, Shariati Hospital, Tehran University of Medical Science, Tehran 14166-34793, Iran

**Keywords:** traumatic brain injuries, diagnostic imaging, computed tomography scan, trauma severity indices, prognosis

## Abstract

This study compared the predictive utility of Marshall, Rotterdam, Stockholm, Helsinki, and NeuroImaging Radiological Interpretation System (NIRIS) scorings based on early non-contrast brain computed tomography (CT) scans in patients with traumatic brain injury (TBI). The area under a receiver operating characteristic curve (AUROC) was used to determine the predictive utility of scoring systems. Subgroup analyses were performed among patients with head AIS scores > 1. A total of 996 patients were included, of whom 786 (78.9%) were males. In-hospital mortality, ICU admission, neurosurgical intervention, and prolonged total hospital length of stay (THLOS) were recorded for 27 (2.7%), 207 (20.8%), 82 (8.2%), and 205 (20.6%) patients, respectively. For predicting in-hospital mortality, all scoring systems had AUROC point estimates above 0.9 and 0.75 among all included patients and patients with head AIS > 1, respectively, without any significant differences. The Marshall and NIRIS scoring systems had higher AUROCs for predicting ICU admission and neurosurgery than the other scoring systems. For predicting THLOS ≥ seven days, although the NIRIS and Marshall scoring systems seemed to have higher AUROC point estimates when all patients were analyzed, five scoring systems performed roughly the same in the head AIS > 1 subgroup.

## 1. Introduction

Traumatic brain injury (TBI), known as the silent epidemic, is a major cause of death and disability worldwide [1,2]. The worldwide incidence and prevalence of TBI have been increasing, as the Global Burden of Disease study estimated that from 1999 to 2016, the age-standardized incidence and prevalence rates of TBI increased by 3.6% and 8.4%, respectively [2].

Trauma scoring systems summarize the severity of injuries sustained by a patient into a single number [3]. However, translating the injuries into a single number may have some disadvantages, such that we might need to disregard some injury details and put patients with varying injuries into a single severity category [3]. Therefore, a sound scoring system should be designed sensibly to differentiate between patients with favorable and unfavorable outcomes [3].

Brain computed tomography (CT) is the imaging modality of choice for the primary evaluation of patients suspected of TBI in emergency departments. Thus far, some scoring systems based on non-contrast brain CT scans have been introduced. The Marshall classification was primarily published in 1991 [4]. This scoring system classifies patients with TBI into six categories, considering the amount of midline shift (MLS), the status of cisterns, whether surgical evacuation was performed, the presence of high- or mixed-density lesions, and whether these density lesions are larger than 25 milliliters (mL) [4]. The first category includes patients with no visible intracranial pathologies, and the sixth category includes patients with lesion densities larger than 25 mL that were not surgically evacuated [4].

In 2005, the Rotterdam score was published, trying to solve some of the shortcomings of the Marshall classification [5]. The Rotterdam score ranges from one to six, similar to the Marshall classification and the motor component of the Glasgow Coma Scale (GCS). In contrast to the Marshall classification, the Rotterdam score considers the presence of subarachnoid hemorrhage (SAH) or intraventricular hemorrhage (IVH) as a particular item in the scoring system. Moreover, the Marshall classification does not distinguish between various types of mass lesions. Therefore, due to the better prognosis of epidural hemorrhage (EDH) compared with intraparenchymal hemorrhage (IPH) and subdural hemorrhage (SDH), developers of the Rotterdam scoring system decided that a point should be added to the total Rotterdam score if EDH is absent (not present) [5]. In contrast to the Marshall classification, the Rotterdam score does not consider the mass lesion size directly as a separate item.

The Stockholm scoring system was published in 2010 [6]. This scoring system has some unique features compared to previous scoring systems. It has a separate subscore named traumatic SAH (tSAH) that consists of SAH depth in convexities, SAH depth in basal cisterns, and the presence of IVH. The tSAH score ranges from zero to six, and after being divided by two, this score is used as a component to calculate the Stockholm tally score. In the Stockholm tally score, the amount of MLS is considered a continuous variable rather than a categorized variable. It also considers diffuse axonal injuries (DAI) in scoring. Regarding the mass lesion types, this scoring system subtracts a point when EDH is present and adds a point in the case of a dual-sided SDH.

In 2014, the Helsinki scoring system was introduced [7]. In contrast to the Marshall scoring system, which does not consider mass lesion type, and the Rotterdam scoring system, which only considers EDH as a distinct item, the Helsinki score distinguishes SDH, intracerebral hemorrhage (ICH), and EDH types of mass lesions. It also considers IVH a separate item. However, in contrast to the Rotterdam score, this scoring system does not consider SAH.

The NeuroImaging Radiological Interpretation System (NIRIS) was introduced in 2018 as an outcome-based scoring system that facilitates clinical decision-making [8]. This scoring system classifies patients with TBI into five distinct categories from zero to four, and a specific clinical decision has been recommended for each type. In 2019, a revised version of this scoring system was introduced [9]. In the revised version, hemorrhage/contusion thresholds were modified; moderate hydrocephalus was moved to the NIRIS 2 category; pneumocephalus was assigned to the NIRIS 1 category; and MLS of more than 10 mm was assigned to the NIRIS 4 category [9].

In this retrospective study, we aimed to compare the predictive utility of scoring systems based on early non-contrast brain CT scans of patients with head injuries who had undergone CT with suspicion of sustaining TBI.

## 2. Materials and Methods

This retrospective study was conducted on patients who had sustained or were suspicious of sustaining TBI and were hospitalized at Shahid Beheshti Hospital in Kashan, Iran, during 2017–2021. Initial identification of patients and extraction of basic data were made through the National Trauma Registry of Iran (NTRI) [10]. We included patients with head injuries for whom at least one non-contrast brain CT scan had been performed. Patients who (i) had left the hospital with personal consent or without notice, (ii) had unknown discharge status, (iii) were transferred from Shahid Beheshti Hospital to other hospitals to continue the treatment process, (iv) had significant brain CT findings of nontraumatic origins such as brain tumors and ischemic stroke, (v) did not have any non-contrast brain CT scans before performing the neurosurgical intervention, (vi) died during the hospitalization, but had no abnormal trauma-related findings in their initial CT scans, and (vii) had major extra-cranial injuries were excluded. Major extra-cranial injuries were defined as injuries to the chest or abdomen with an Abbreviated Injury Scale (AIS) score > 3 [11,12].

For patients with more than one non-contrast brain CT scan, the first CT scan performed after the hospital admission was used for scoring. All brain CT measurements were completed using the picture archival communication system (PACS). The files used for scoring were in Digital Imaging and Communications in Medicine (DICOM) format. To calculate the brain hematoma size, we used the ABC/2 formula, in which the A and B parameters refer to the maximal length and width of the hematoma in the axial plane, respectively. The A and B lines were chosen to be perpendicular to each other. To calculate the C parameter, the number of axial slices in which the hematoma could be observed was multiplied by the thickness of the CT slices. To calculate the MLS, first the most anterior and posterior visible points on the falx cerebri were connected using a line. Then, the farthest distance between this line and the septum pellucidum was recorded as the MLS. At least two radiologists reviewed the scans. An independent board-certified radiologist resolved any disagreement with the CT findings.

In order to be included in our study, patients who had undergone neurosurgery had to have at least one non-contrast brain CT scan prior to the surgery. To simulate the real-world setting and since neurosurgery was one of the outcome variables in our study, we did not assign any patient based on their first post-admission CT to the Marshall 5 category, i.e., the category that includes patients with any surgically evacuated lesions. Outcome variables were discharge status (in-hospital mortality), ICU admission, need for neurosurgery (craniotomy/craniectomy, burr hole, ventricular drain, and duraplasty), and total hospital length of stay (THLOS). In our study, the third quartile (Q3) of THLOS was six days, and we defined prolonged hospitalization as THLOS of seven days or more. Abujaber et al. [13] also used a similar approach to determine the prolonged length of stay in a previous study. Performing cranial osteoplasty alone was not considered a neurosurgical intervention.

### Statistical Analysis

Data were analyzed using Stata software version 14.0 (Stata Corp, College Station, TX, USA). The Mann–Whitney U test was used to compare the brain CT scores between outcome subgroups. The chi-squared test or Fisher’s exact test was used to compare the distribution of patients in the Marshall, Rotterdam, NIRIS, and Helsinki categories according to outcome variables. Then, the area under a receiver operating characteristic curve (AUROC) was used to determine the predictive utility of CT scan-based scoring systems. For testing the equality of AUROCs, Stata’s roccomp command was used. Subgroup analyses were performed among patients who had a head AIS score > 1 based on the first non-contrast brain CT scan. AIS > 1 cut-off has also been used in previous studies. For instance, while designing the Predictor of Isolated Trauma in the Head (PITH) scoring system, Lee et al. [14] used AIS > 1 for dichotomization, i.e., they coded head AIS = 1 as 1 and head AIS > 1 as 2 in their suggested predictive model. Moreover, the AIS codes of some intracranial injuries, such as SAH not further specified (NFS) and IVH NFS, have a severity score of 2 in the AIS dictionary [12]. Therefore, these codes would have been excluded from our subgroup analysis if we had chosen AIS > 2 as the cut-off value. Hence, we decided that AIS > 1 would be more suitable for subgroup analysis in our study. The weighted kappa statistic assessed the inter-rater agreements between the two independent raters.

This study was approved by the Ethics Committee of Sina Hospital, Tehran University of Medical Sciences (IR.TUMS.SINAHOSPITAL.REC.1399.103).

## 3. Results

A total of 996 patients were included in the study. Patients’ ages ranged from 0 (less than one year) to 95 years, with an average (±SD) of 38.8 (±21.1) years. Road traffic incidents were the most common cause of injury (n = 750, 75.3%), followed by falls (n = 195, 19.6%). Among the patients who were injured due to falls, 65 (33.3%) and 121 (62.1%) patients were injured due to falls from standing height (i.e., had a fall height of zero) and falls from height, respectively. Nine (4.6%) patients had unknown fall heights. The majority (87.0%) of patients in our study had an admission GCS score of 13–15. In-hospital mortality, ICU admission, neurosurgery, and THLOS ≥ seven days were recorded for 27 (2.7%), 207 (20.8%), 82 (8.2%), and 205 (20.6%) patients, respectively. The median (interquartile range; IQR) of ICU length of stay (LOS) and THLOS were 4 (9) and 3 (4) days, respectively. Patients’ baseline characteristics, outcome variables, and CT findings are presented in Table 1. Table 2 shows positive CT findings in TBI deaths according to the variables of five scoring systems. Inter-rater agreements were substantial for all CT scoring systems (0.74 to 0.80).

Based on the non-parametric comparison of brain CT scores, the differences observed between the outcome subgroups of all four outcomes (in-hospital mortality, ICU admission, neurosurgery, and THLOS of seven days or more) were found to be statistically significant (*p* value < 0.001). The median (IQR) of brain CT scores according to in-hospital mortality, ICU admission, neurosurgery, and THLOS ≥ seven days outcomes are shown in Table 3. The distributions of patients in the Marshall, Rotterdam, NIRIS, and Helsinki categories according to outcome variables have been provided in a Appendix A.

The AUROCs (95% CI) of brain CT scores for predicting the studied outcomes are shown in Table 4. All five scoring systems had AUROC point estimates above 0.9 in predicting in-hospital mortality, and there were no significant differences among their AUROCs in this regard. Marshall and NIRIS had higher AUROCs than other scoring systems in predicting ICU admission and neurosurgery outcomes. In predicting THLOS ≥ seven days, the Marshall and NIRIS scoring systems had higher AUROC point estimates; however, differences among the AUROCs were less pronounced, and some overlaps were observed among 95% CIs. Figure 1 shows the receiver operating characteristic (ROC) curves of scoring systems for predicting the studied outcomes.

In the subgroup analysis of patients with head AIS > 1, scoring systems had point estimates above 0.75 for predicting in-hospital mortality, and according to the overlap among 95% CIs, observed differences were concluded to be insignificant. The NIRIS and Marshall scoring systems performed better in predicting ICU admission and neurosurgery; however, there were minor overlaps among the 95% CIs of the scoring systems for predicting ICU admission. There were no significant differences among the scoring systems in predicting THLOS ≥ seven days. The AUROC (95% CI) of scoring systems in the subgroup of patients with head AIS > 1 is presented in Table 5.

## 4. Discussion

In this study, we compared the utility of scoring systems based on early non-contrast brain CT scans in predicting outcomes of in-hospital mortality, ICU admission, neurosurgery, and THLOS ≥ seven days among patients who had sustained or were suspicious of sustaining TBI and a subgroup of patients who had head AIS > 1 based on their first non-contrast brain CT scans. For predicting in-hospital mortality, the scoring systems had AUROC point estimates above 0.9 and 0.75 among all included patients and patients with head AIS > 1, respectively. 

The Marshall and NIRIS scoring systems had higher AUROCs for predicting ICU admission and neurosurgery than the other scoring systems. For predicting THLOS ≥ seven days, although the NIRIS and Marshall scoring systems seemed to have higher AUROC point estimates when all patients were analyzed, the scoring systems performed roughly the same in the head AIS > 1 subgroup. To interpret these observations, the similarities and differences between the scoring systems should be scrutinized. 

One of these differences is how EDH is scored in each scoring system. EDH has the best prognosis among extra-axial hematomas, and the mortality rate following EDH is lower than that of SDH [15,16]. Therefore, the Rotterdam scoring system was designed so that in the absence (non-presence) of EDH on a patient’s brain CT scan, one point is added to the total severity score [5]. In the Helsinki and Stockholm scoring systems, if EDH is observed on a patient’s brain CT scan, three points and one point are subtracted from the patient’s total severity score, respectively [6,7]. On the other hand, the Marshall scoring system seems to adopt a neutral position since it does not differentiate among various types of intracranial hematoma. In the NIRIS scoring system, the presence of EDH per se does not lead to a lower total score. This approach may be a reason for the poorer performance of the Rotterdam, Helsinki, and Stockholm scoring systems in predicting ICU admission and neurosurgical procedures compared with the NIRIS and Marshall scoring systems. A question might arise: if the NIRIS scoring system considers EDH an adverse prognostic factor even though EDH has a lower mortality rate than other extra-axial hematomas, why does it perform similarly to other scoring systems in predicting in-hospital mortality? The reason might be that the NIRIS takes more details into account. For instance, duret hemorrhage is associated with a high rate of mortality [17], and the NIRIS is the only scoring system that takes duret hemorrhage into account and assigns it to the most severe scoring category, i.e., the NIRIS 4. On the other hand, EDH could not always be a favorable prognostic indicator. Sometimes the presence of EDH with DAI or high-volume EDH causing a midline shift and brain herniation could worsen the outcome. These details may be better to consider in the scoring systems [18].

Another difference among scoring systems is the way they consider cerebral contusions in scoring. The NIRIS scoring system directly considers cerebral contusions. As mentioned, the Marshall score adopts a neutral position and does not differentiate among various lesion densities. To our knowledge, the Stockholm and Rotterdam systems do not consider cerebral contusions directly in scoring. The Helsinki scoring system has a different item for intracerebral hematomas. However, it has not made a clear distinction between contusions and intracerebral hematomas, which may lead to varying interpretations regarding whether or not to consider contusions of various sizes in scoring. This issue stems from the vague distinction between ICH/IPH and cerebral contusion, and some arbitrary cutoffs have even been proposed [19]. On the other hand, the NIRIS scoring system has differentiated contusions and IPH as two different scoring items. With all that being said, the way that cerebral contusions are considered in scoring might be another reason for the better performance of the NIRIS and Marshall systems in our study. We recommend that scoring system designers take a clear position on this issue and provide a detailed manual for their scoring systems.

It should also be noted that predicting patients’ outcomes solely based on brain CT findings might be erroneous. For instance, previous studies have found that patients’ baseline characteristics, such as age, and their functional status, such as the GCS score, are associated with their outcomes following TBI [20,21,22,23]. In other words, patients with similar brain CT scores but different baseline characteristics and functional scores might have different outcomes. On the other hand, although non-contrast brain CT is considered the modality of choice for a rapid initial evaluation of patients with TBI, it has some limitations, such as lower detection and underestimation of DAI and parenchymal contusions compared with the more advanced imaging modalities [24].

Another important point to note is that we used the first post-admission brain CT scan for patients’ severity scoring. According to previous studies, using the initial CT scan, also known as the admission CT scan, might pose some problems. For instance, Nagesh et al. observed that among patients with mild/moderate TBI and abnormal initial CT scans who had not undergone neurosurgical intervention, some might demonstrate a progression of existing lesions or the evolution of new lesions on repeat CT scans [25]. They also reported that some patients eventually underwent neurosurgical intervention due to the exacerbation of repeat CT scans without neurological worsening [25]. This indicates that the repeat CT rather than the admission CT may be used as a basis for clinical decision-making. Servadei et al. also emphasized the importance of repeat CT scans and considering the worst rather than the first CT scan to predict the outcomes of patients with TBI [26]. Stocchetti et al. investigated the time course of raised intracranial pressure (RICP) and reported that the evolution of RICP can be delayed in many patients [27]. Iaccarino et al. concluded that among patients with cerebral contusions, the development of or increase in MLS in follow-up CT scans is associated with deterioration of patients’ status during the initial hours post-injury and an unfavorable outcome [28]. Some previous studies also used initial head CT for scoring, such as the study conducted by Thelin et al. in Sweden and Finland [29]. Thelin et al. recommended that in future studies, researchers should determine which time point has the highest utility for predicting patient outcomes [29]. We believe that although using the worst head CT might be more reasonable for outcome prediction, physicians, especially front-line physicians managing patients in emergency departments, might wonder about the comparative performance of these scoring systems based on the initial head CT scans.

According to Nagesh et al. [25] and Servadei et al. [26], some imaging and non-imaging clues on admission might predict the worsening of repeat CT scans. Therefore, a scoring system that emphasizes these clues can be utilized to predict patients’ outcomes during the initial hours of admission.

In a study by Munakomi, the utility of the Marshall and Rotterdam scoring systems for predicting the outcome of patients with TBI was investigated [30]. The Rotterdam and Marshall scores were calculated based on the initial CT scan, and the investigated outcome was early death, which was measured based on the patients’ discharge status. Munakomi reported the AUROCs of the Marshall and Rotterdam scoring systems to be 0.912 and 0.929, respectively [30]. In a study by Deepika et al., the AUROCs of the Marshall and Rotterdam scoring systems for predicting two-week mortality were reported to be 0.707 and 0.681, respectively [31]. Vehviläinen et al. concluded that the Helsinki score better predicted six-month mortality than the NIRIS scoring system [32]. Thelin et al. concluded that the Helsinki and Stockholm scoring systems performed better than the Marshall and Rotterdam systems in predicting the long-term unfavorable outcome measured by the Glasgow Outcome Scale (GOS) [29]. It should be noted that the TBI severity distribution and the way previous studies defined inclusion criteria might play a role in the observed discrepancies. In Deepika et al.’s study, only patients with moderate and severe injuries were included; the study by Vehviläinen et al. was conducted on ICU-admitted patients; Thelin et al. included patients who required neuro-intensive care; and Munakomi included patients with mild to severe injuries. In our study, we even went one step further and included head-injured patients that had undergone brain CT, indicating the suspicion of sustaining TBI. A similar approach, i.e., including patients suspicious of sustaining TBI, was also adopted by Wintermark et al. [8]. With all that being said, it can be concluded that scoring systems may have different predictive performances in different subgroups of TBI, and we suggest putting more emphasis on this issue in future studies.

Another important issue is that in cases where scoring systems have relatively similar AUROCs for predicting a specific outcome measure, other aspects of these systems, such as user-friendliness and the amount of time required for filling them out, play a major role in choosing between them. For instance, Creeden et al. mentioned that the NIRIS was the most preferred scoring system by image reviewers in their study [33]. Therefore, for future studies, we also suggest placing more emphasis on assessing these aspects of scoring systems.

### Limitations

This study was retrospective, and the initial identification of patients and basic data extraction were made through the NTRI. However, future studies are warranted to assess this registry program’s patient coverage and data quality. Most patients included in our study had GCS scores of 13–15, and only 20.8% were admitted to the ICU. Therefore, this issue is a limitation of our research since many of the brain CT scoring systems may have been designed for severe TBI patients requiring ICU admission. However, from another perspective, it may emphasize the necessity of developing new scoring systems or recognizing scoring systems that have a higher utility for patients with non-severe injuries and patients who are suspicious of sustaining TBI. Moreover, the GCS scores in the NTRI database were collected mostly retrospectively through patient files, and the worst GCS scores were unavailable.

Another limitation is that our study included both isolated and multiple trauma patients. In other words, patients with injuries to body regions other than the head were also included in this study. This is consistent with the fact that road traffic incidents were our study’s most common cause of injury. Therefore, we believe that stratifying the results based on isolated and multiple trauma patients will be of benefit in future studies.

## 5. Conclusions

The Marshall, Rotterdam, Helsinki, Stockholm, and NIRIS scoring systems had high utility in predicting in-hospital mortality, and there were no significant differences among their AUROCs. The Marshall and NIRIS scoring systems had higher AUROCs for predicting ICU admission and neurosurgery than the other scoring systems. For predicting THLOS ≥ seven days, although the NIRIS and Marshall scoring systems seemed to have higher AUROC point estimates when all patients were analyzed, the scoring systems performed roughly the same in the head AIS > 1 subgroup.

## Figures and Tables

**Figure 1 brainsci-13-01145-f001:**
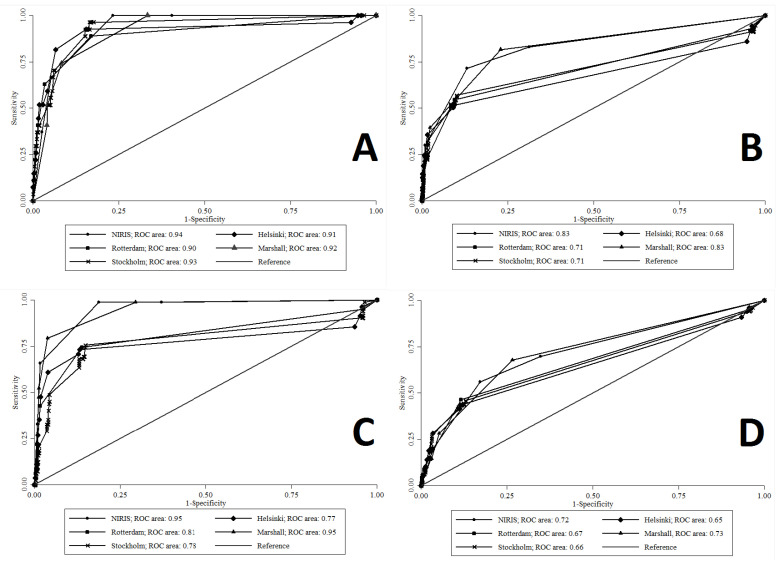
Comparison of results of ROC analysis for prediction of (**A**) mortality, (**B**) ICU admission, (**C**) neurosurgical procedure, and (**D**) prolonged hospitalization by the Marshall, Rotterdam, Stockholm, Helsinki, and NIRIS scoring systems.

**Table 1 brainsci-13-01145-t001:** Distribution of baseline characteristics, outcome variables, and CT findings.

		Number (%)
Gender	Male	786 (78.9)
	Female	210 (21.1)
Age (years)	0–14	110 (11.0)
	15–44	506 (50.8)
	45–64	236 (23.7)
	≥65	144 (14.5)
GCS score	13–15	866 (87.0)
	9–12	65 (6.5)
	3–8	59 (5.9)
Outcomes	ICU Admission	207 (20.8)
	THLOS ≥ Seven Days	205 (20.6)
	Neurosurgery	82 (8.2)
	In-hospital Mortality	27 (2.7)
CT findings	Skull Fracture	216 (21.6)
	SAH	163 (16.3)
	SDH	126 (12.6)
	Parenchymal Contusion	126 (12.6)
	Abnormal Cisterns	107 (10.7)
	EDH	79 (7.9)
	Pneumocephalus	73 (7.3)
	Midline Shift	45 (4.5)
	IPH/ICH	41 (4.1)
	Mass Lesion > 25 cm3	31 (3.1)
	Focal Herniation	27 (2.7)
	IVH	24 (2.4)
	Hydrocephalus	5 (0.5)
	Duret Hemorrhage	4 (0.4)
	DAI	2 (0.2)
	Surgical Evacuation	0 ^1^

^1^ The first CT scan after the hospital admission was used for scoring. CT: computed tomography; DAI: diffuse axonal injury; EDH: epidural hematoma; GCS: Glasgow Coma Scale; ICH: intracerebral hematoma; ICU: intensive care unit; IPH: intraparenchymal hemorrhage; IVH: intraventricular hemorrhage; SAH: subarachnoid hemorrhage; SDH: subdural hematoma; TBI: traumatic brain injuries; THLOS: total hospital length of stay.

**Table 2 brainsci-13-01145-t002:** Positive CT findings according to the variables of scoring systems in 27 dead TBI patients, sorted by frequency.

	Marshal	Rotterdam	Helsinki	Stockholm	NIRIS
SAH	-	22	-	22	22
SDH	-	-	16	8 ^1^	16
Skull Fracture	-	-	-	-	13
Abnormal Cisterns	11	11	11	-	-
Mass Lesion > 25 cm^3^	11	-	11	-	-
Focal Herniation	-	-	-	-	11
IPH/ICH	-	-	8	-	8
Parenchymal Contusion	-	-	-	-	8
IVH	-	7	7	7	7
Midline Shift (>5 mm)	5	5	-	5	5
Pneumocephalus	-	-	-	-	4
EDH ^2^	-	3	3	3	3
Hydrocephalus	-	-	-	-	3
Duret Hemorrhage	-	-	-	-	2
DAI	-	-	-	1	1
Surgical Evacuation	0	-	-	-	-

^1^ Dual-sided SDH. ^2^ EDH is calculated as a positive prognostic factor in the Rotterdam, Helsinki, and Stockholm systems. CT: computed tomography; DAI: diffuse axonal injury; EDH: epidural hematoma; ICH: intracerebral hematoma; IPH: intraparenchymal hemorrhage; IVH: intraventricular hemorrhage; mm: millimeter; NIRIS: NeuroImaging Radiological Interpretation System; SAH: subarachnoid hemorrhage; SDH: subdural hematoma; TBI: traumatic brain injuries.

**Table 3 brainsci-13-01145-t003:** Brain CT scores according to in-hospital mortality, ICU admission, neurosurgery, and THLOS ≥ seven days.

Scoring Systems	In-Hospital Mortality (n = 996)			ICU Admission (n = 994) ^1^			Neurosurgery (n = 996)			THLOS ≥ Seven Days (n = 995) ^2^		
	Yes	No	*p*	Yes	No	*p*	Yes	No	*p*	Yes	No	*p*
Marshall, median (IQR)	4 (4)	1 (1)	<0.001	2 (1)	1 (0)	<0.001	4 (3)	1 (1)	<0.001	2 (2)	1 (1)	<0.001
Rotterdam, median (IQR)	4 (2)	2 (0)	<0.001	3 (1)	2 (0)	<0.001	3 (2)	2 (0)	<0.001	2 (1)	2 (0)	<0.001
Helsinki, median (IQR)	6 (5)	0 (0)	<0.001	2 (3)	0 (0)	<0.001	3 (6)	0 (0)	<0.001	0 (3)	0 (0)	<0.001
Stockholm, median (IQR) ^3^	3.00 (2.00)	1.00 (0)	<0.001	2.00 (1.45)	1.00 (0)	<0.001	2.00 (1.92)	1.00 (0)	<0.001	1.00 (1.15)	1.00 (0)	<0.001
NIRIS, median (IQR)	3 (2)	0 (1)	<0.001	2 (2)	0 (1)	<0.001	3 (2)	0 (1)	<0.001	2 (2)	0 (1)	<0.001

^1^. Two patients had unknown ICU admission status. ^2^. One patient had an unknown THLOS. ^3^. Stockholm score is a continuous variable. Therefore, two decimal places were reported. CT: computed tomography; ICU: intensive care unit; IQR: interquartile range; NIRIS: NeuroImaging Radiological Interpretation System; THLOS: total hospital length of stay.

**Table 4 brainsci-13-01145-t004:** AUROCs (95% CI) of brain CT scores for predicting in-hospital mortality, ICU admission, neurosurgery, and THLOS ≥ seven days (all patients).

Scoring Systems	Outcome Variables			
	In-Hospital Mortality (n = 996)	ICU Admission (n = 994) ^1^	Neurosurgery (n = 996)	THLOS ≥ Seven Days (n = 995) ^2^
Marshall	0.92 (0.89–0.95)	0.83 (0.80–0.86)	0.95 (0.93–0.97)	0.73 (0.69–0.76)
Rotterdam	0.90 (0.84–0.97)	0.71 (0.67–0.75)	0.81 (0.75–0.87)	0.67 (0.63–0.71)
Helsinki	0.91 (0.84–0.99)	0.68 (0.63–0.72)	0.78 (0.70–0.85)	0.65 (0.60–0.69)
Stockholm	0.93 (0.89–0.97)	0.71 (0.67–0.76)	0.78 (0.72–0.85)	0.66 (0.62–0.70)
NIRIS	0.94 (0.91–0.96)	0.83 (0.80–0.87)	0.95 (0.93–0.97)	0.72 (0.68–0.76)

^1^. Two patients had unknown ICU admission status. ^2^. One patient had an unknown THLOS. AUROC: area under a receiver operating characteristic curve; CI: confidence interval; CT: computed tomography; ICU: intensive care unit; NIRIS: NeuroImaging Radiological Interpretation System; THLOS: total hospital length of stay.

**Table 5 brainsci-13-01145-t005:** AUROCs (95% CI) of brain CT scores for predicting in-hospital mortality, ICU admission, neurosurgery, and THLOS ≥ seven days in the subgroup of patients with head AIS > 1.

Scoring Systems	Outcome Variables			
	In-Hospital Mortality (n = 419)	ICU Admission (n = 418) ^1^	Neurosurgery (n = 419)	THLOS ≥ Seven Days (n = 418) ^2^
Marshall	0.80 (0.72–0.88)	0.77 (0.73–0.81)	0.88 (0.84–0.92)	0.69 (0.64–0.73)
Rotterdam	0.84 (0.76–0.92)	0.69 (0.64–0.74)	0.75 (0.69–0.81)	0.68 (0.63–0.73)
Helsinki	0.87 (0.79–0.94)	0.69 (0.64–0.74)	0.75 (0.68–0.82)	0.68 (0.62–0.73)
Stockholm	0.76 (0.67–0.85)	0.70 (0.65–0.75)	0.67 (0.61–0.74)	0.68 (0.63–0.73)
NIRIS	0.84 (0.78–0.90)	0.78 (0.74–0.82)	0.89 (0.86–0.92)	0.69 (0.64–0.73)

^1^. One patient had an unknown ICU admission status. ^2^. One patient had an unknown THLOS. AUROC: area under a receiver operating characteristic curve; CI: confidence interval; CT: computed tomography; ICU: intensive care unit; NIRIS: NeuroImaging Radiological Interpretation System; THLOS: total hospital length of stay.

## Data Availability

The data presented in this study are available on request from the corresponding author.

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
