# Peer review of "Comparing Predictive Utility of Head Computed Tomography Scan-Based Scoring Systems for Traumatic Brain Injury: A Retrospective Study"

_brainsci, 2023, doi:10.3390/brainsci13081145_

Round 1

Reviewer 1 Report

Dear authors. Congratulations on the interesting study. Well designed and well-performed. My only consideration is regarding the AIS selected for subanalysis. There is not a clear explanation of why select this AIS range. In general, the AIS >2 is the selected AIS, especially for understanding the behavior of more complicated cases (moderate to severe). A better explanation of why was selected this AIS score for the subanalysis will be ideal. 

Author Response

We sincerely appreciate your valuable comments. As you have mentioned, AIS > 2 might be used more often by researchers for dichotomizing the patients into more severely and less severely injured categories. Although less often, AIS > 1 has also been used in previous studies. For instance, while designing the Predictor of Isolated Trauma in Head (PITH) scoring system, Lee et al. (1) used AIS > 1 for dichotomization. Moreover, AIS codes of some intracranial injuries such as subarachnoid hemorrhage (SAH) not further specified (NFS) (AIS code 140693) and intraventricular hemorrhage (IVH) NFS (AIS code 140678) have severity score of 2 in the AIS dictionary; therefore, these codes would have been excluded from our subgroup analysis if we had chosen AIS > 2 as the cut-off value, and it was not something that we wanted, because the way SAH and IVH are considered in severity scoring is a distinguishing feature of each scoring system. Hence, we decided that AIS > 1 would be more suitable for subgroup analysis in our study. We added the following explanation to the Methods section:

“AIS > 1 cut-off has also been used in previous studies. For instance, while designing the Predictor of Isolated Trauma in Head (PITH) scoring system, Lee et al. (14) used AIS > 1 for dichotomization. Moreover, AIS codes of some intracranial injuries such as SAH not further specified (NFS) and IVH NFS have a severity score of 2 in the AIS dictionary (12). Therefore, these codes would have been excluded from our subgroup analysis if we had chosen AIS > 2 as the cut-off value. Hence, we decided that AIS > 1 would be more suitable for subgroup analysis in our study.”

Reviewer 2 Report

An original research paper on the comparison of the power of different CT severity scoring systems in patients with head injury on admission to hospital was submitted for review. The authors compared Marshall, Rotterdam, Stockholm, Helsinki and NIRIS. The article is relevant and of unquestionable scientific interest. The merit of the article lies in the large sample of tomograms on which the study was carried out, almost 1000 people.

The authors carried out the necessary statistical calculations, which showed that all the scales were similar in predicting outcomes and length of hospital stay. 

There are many disadvantages and limitations of the study. The authors have listed them all in a separate section.

In my opinion, the article is sufficiently illustrated, holistically designed, and can be published as presented.

Author Response

Thank you for the careful review of our manuscript. 

Reviewer 3 Report

This is an interesting paper comparing different CT scoring systems for determining the outcomes of Traumatic Brain Injured patients . Unfortunately there are some major issues to be addressed by the authors 

1) The Marshall scoring system (the first ever published) is coming from a Traumatic COMA data bank . It means that all patients were severe head injuries . The European Brain Injury Consortium data base contained at least 50% of severely injured patients .The Stockholm and Helsinki score were applied to patients admitted to ICU , the Rotterdam score originally to a majority of  severely head injured patents  The casistics used by the authors to validate the different scoring systems are constituted by 87% of mild head injured patients... 

2) In this scenario the outcome on discharge, the mortality and the more severe CT data are limited to a very small subpopulation 

3) As it was already published the first CT rarely collect the real anatomical post traumatic damage : the outcome is more related to the worst than to the initial CT scan . Nevertheless in countries with limited facilities only one CT scan is often allowed and therefore these data may be of interest 

This work is well organized and interesting I suggest the authors to use only mild injuries with complication (evacuated hematoma or so) , moderate and severe head injures patients 

ok

Author Response

Thank you for your insightful comments.

Regarding comments #1 & #2: As you have correctly mentioned, these scoring systems were designed using data from patient populations with varying severities and characteristics. However, researchers and clinicians might still be curious about the performance of these scoring systems among patient populations different from those that scoring systems were originally designed for. Hence, there might not be a single best answer to the question “Which head CT scan-based scoring system is the best of all for predicting TBI patients’ outcomes?” because the answer might vary according to factors such as the studied population characteristics. In our study, we assessed the predictive utilities of these scoring systems among a specific patient population, and we do not intend to generalize our results to patient populations different from ours. We also encouraged researchers to assess the predictive utilities of these scoring systems among other patient populations in future studies.

Regarding comment #3: We agree with you that assessing the comparative performance of scoring systems based on the initial head CT would be of great interest to resource-limited countries, but we also believe that it would be of interest to other countries, too. In a study conducted in Sweden and Finland, Thelin et al. used patients’ initial head CT scans to calculate the Marshall, Rotterdam, Helsinki, and Stockholm scores, and they recommended that in future studies, researchers should determine which time point has the highest utility for predicting patient outcomes (2). We believe that although using the worst head CT would be more reasonable for outcome prediction, physicians, especially front-line physicians managing patients in emergency departments might wonder about the comparative performance of these scoring systems based on the initial head CT scans. We added the following explanation to the Discussion section:

“Some previous studies also used initial head CT for scoring, such as the study conducted by Thelin et al. in Sweden and Finland (28). Thelin et al. recommended that in future studies, researchers should determine which time point has the highest utility for predicting patient outcomes (28). We believe that although using the worst head CT might be more reasonable for outcome prediction, physicians, especially front-line physicians managing patients in emergency departments might wonder about the comparative performance of these scoring systems based on the initial head CT scans.”

Regarding your suggestion for using only mild injuries with complications, moderate, and severe injuries, we believe that subgroup analysis based on head AIS > 1 excludes patients with uncomplicated mild injuries to a large extent, because head AIS ≤ 1 generally involves patients who were suspicious of sustaining TBI, but had normal initial head CT scans and patients who were less severely injured, such as those with scalp injuries. Although some other criteria could also be thought of for subgroup analysis, we thought that head AIS > 1 would be a more suitable one for our study.

Round 2

Reviewer 3 Report

The authors have sufficiently replayed to my suggestions 

no